# The Eyes Absent Proteins: Unusual HAD Family Tyrosine Phosphatases

**DOI:** 10.3390/ijms22083925

**Published:** 2021-04-10

**Authors:** Kaushik Roychoudhury, Rashmi S. Hegde

**Affiliations:** Division of Developmental Biology, Cincinnati Children’s Hospital Medical Center, Department of Pediatrics, University of Cincinnati College of Medicine, 3333 Burnet Avenue, Cincinnati, OH 45229, USA; kaushik.roychoudhury@cchmc.org

**Keywords:** HAD, eyes absent, EYA1, EYA2, EYA3, EYA4, EYA, tyrosine phosphatase, PTP

## Abstract

Here, we review the haloacid dehalogenase (HAD) class of protein phosphatases, with a particular emphasis on an unusual group of enzymes, the eyes absent (EYA) family. EYA proteins have the unique distinction of being structurally and mechanistically classified as HAD enzymes, yet, unlike other HAD phosphatases, they are protein tyrosine phosphatases (PTPs). Further, the EYA proteins are unique among the 107 classical PTPs in the human genome because they do not use a Cysteine residue as a nucleophile in the dephosphorylation reaction. We will provide an overview of HAD phosphatase structure-function, describe unique features of the EYA family and their tyrosine phosphatase activity, provide a brief summary of the known substrates and cellular functions of the EYA proteins, and speculate about the evolutionary origins of the EYA family of proteins.

## 1. The Haloacid Dehalogenase (HAD) Family Phosphatases

Hydrolases, enzymes that utilize water to catalyze the breakdown of chemical bonds, are sub-divided based upon the specific bond that is cleaved. Hydrolases that cleave ester bonds (including nucleases, phosphodiesterases, lipases, and phosphatases) form sub-class EC 3.1 in the Enzyme Commission (EC) classification. Among these, the HAD superfamily includes a diverse range of hydrolases including phosphoesterases, dehalogenases, phosphonatases, and sugar phosphomutases. They are characterized by the use of an Aspartate as a nucleophile in a two-step, divalent cation-dependent catalytic reaction. Evolutionarily, HAD enzymes represent one of the earliest known enzyme families. Estimates based on orthologous sequence analyses predict HAD enzymes were already present in the last common ancestor between prokaryotes, eukaryotes, and archaea [1]. The core catalytic domain of HAD enzymes adopts a Rossmannoid a/β fold (a three-layered α/β sandwich made up of repeating β-α units), with a six-residue “squiggle” helical turn just after strand S1, followed by a β-hairpin commonly referred to as the “flap” (Figure 1a). The central β-sheet in the α/β sandwich is made up of parallel strands in 54,123 order. The active site of HAD phosphatases is at the C-terminal end of the β-sheet. Four conserved sequence motifs characterize the HAD family (Figure 1b): a DXDxT/V (motif 1) at the end of strand S1, a serine or threonine residue at the end of strand S2 (motif II), a conserved lysine residue near the N-terminus of the α-helix upstream of strand S4 (Motif III), and conserved acidic residues at the end of strand S4 (Motif IV). While motifs II, III, and IV stabilize reaction intermediates, motif I directly participates in the catalytic reaction, and the acidic residues in motif IV co-ordinate the Mg ion in the active site (Figure 1c,d). The HAD phosphatases utilize a two-step reaction mechanism. In the first step, the first Asp residue in motif I acts as a nucleophile attacking the phosphoryl group of the substrate and forming a covalent phosphoaspartyl intermediate (Figure 1d). In the next step, the second Asp acts as a general acid-base deprotonating a water molecule, which then mediates a nucleophilic attack on the phosphoaspartyl intermediate, releasing a free phosphate and regenerating the nucleophilic Asp. The Mg^2+^ cofactor correctly positions the substrate phosphoryl group and acts as an electrostatic shield between the anionic nucleophilic Asp and the di-anionic phosphorylated substrate (Figure 1c) and stabilizes the transition state.

Domains inserted into the flap region or immediately after strand S3 are termed “caps” and serve to both shield the active site and provide a substrate binding cleft. Three classes of caps have been described: C0 caps are typically very small inserts, C1 caps occur in the middle of the β-hairpin flap and form a distinct structural unit (could be α-helical or have an α+β fold as in the P-type ATPases), and C2 caps occur after strand S3. Some HAD family members have both C1 and C2 caps (e.g., phosphoserine phosphatases), and there are instances of C0 family members with an additional C2 cap (e.g., CTD phosphatases). Cap domains serve an essential role in the catalytic activity and substrate specificity of HAD enzymes. Indeed, it has been suggested that this arrangement of an active site between a conserved catalytic core domain and diverse cap domains could contribute to the notable ability of HAD-family enzymes to explore a wide range of substrates over evolutionary time [1].

HAD substrates vary in size and chemistry, ranging from sugars and small metabolites to proteins. There are 40 HAD phosphatase genes in the human genome with the potential to encode 161 proteins [2]. While it is currently unknown how many of these are functional enzymes, HAD phosphatases could account for nearly 23% of all human phosphatase catalytic subunits [2]. The prevailing paradigm holds that the substrates of capped HAD phosphatases tend to be small metabolites that can be trapped in the active site by closure of the cap domain. Typically, macromolecular substrates are the targets of C0-type HAD phosphatases with an open, easily accessible active site (e.g., MDP1, PNKP). However, some capped HAD phosphatases can target both small metabolites as well as protein substrates (chronophin, phosphoglycolate phosphatase), speaking to the high degree of substrate promiscuity permitted by the HAD phosphatase enzyme architecture.

Also notable is the acquisition of a multi-domain architecture in eukaryotic HAD phosphatases, often resulting in proteins with multiple biochemical functions. For instance, soluble epoxide hydrolase 2 (sEH2) has a C-terminal epoxide hydrolase domain and an independent N-terminal HAD phosphatase domain [3,4], and polynucleotide 5′-kinase/3′-phosphatase (PNKP) combines an N-terminal fork-head associated domain with a C-terminal consisting of fused HAD phosphatase and kinase domains [5].

## 2. Eyes Absent (EYA)—A Unique HAD Phosphatase

Unique among the HAD phosphatases are the eyes absent proteins; they are multi-domain proteins and represent the only validated instance of tyrosine phosphatase activity housed in a HAD phosphatase domain [4,5,6,7,8]. The four human EYA paralogs (EYA1-4) are part of a conserved cell-fate determination cascade that was originally described in the context of Drosophila eye development (the retinal determination gene network) [9]. The EYA proteins have a highly conserved C-terminal domain of approximately 270 amino acids that houses all of the motifs characteristic of the HAD phosphatases. This conserved domain is commonly referred to as the EYA domain (ED) (Figure 2a). In addition, a poorly conserved N-terminal domain in the animal EYA proteins can mediate transcriptional activation when tethered to DNA through interaction with the SIX (sine oculis) family of homeodomain proteins. The N-terminal domain also mediates threonine-phosphatase activity, although whether this is intrinsic [10,11,12,13] or via interaction with PP2A [14] remains a matter of discussion. The N-terminal domain has no recognizable structural or functional sequence motifs. Interestingly, plant EYA proteins lack the N-terminal domain and are single-domain tyrosine phosphatase of the HAD class (Figure 2a).

The crystal structure of the catalytic C-terminal domain of EYA2 (EYA2-ED) has been determined [15] (Figure 3). The EYA2-ED domain includes the prototypical HAD Rossmannoid three-layered α/β sandwich comprised of a parallel five-stranded β-sheet sandwiched between four helices. Distinguishing the EYAs from other HAD phosphatases is a large helical C2 “cap” domain inserted after strand S1, and a helix between strands S2 and S3 that protrudes into the CAP domain. Residues that comprise the HAD conserved motifs are at the interface of the core and cap domains and form the substrate-binding site. The Mg^2+^ cofactor is poised to allow the Asp nucleophile to approach and electrostatically stabilize the substrate phosphate group. The cap domain allows for solvent exclusion. In typical HAD reactions, there is movement of the cap domain between “closed” and “open” conformations. No structural evidence for such domain movement is yet available for the EYA proteins. Furthermore, the cap domain of EYA forms a seven-helical bundle that is structurally distinct from the cap domains of all other HAD phosphatases. The helix between strands S2 and S3 that is inserted into this cap domain likely contributes to a degree of rigidity in the relative positions of the cap and core domains.

In addition to being an unusual HAD family phosphatase, the EYA ED domain is also a unique tyrosine phosphatase. All classical protein tyrosine phosphatases (PTPs) are defined by the signature CxxxxxR motif that forms the phosphate binding P-loop in the active site [16]. The nucleophilic Cys and the phosphate-binding Arg residue form a binding pocket for the substrate phosphate group. Catalysis goes through a cysteinyl–phosphate reaction intermediate. The conserved ~280 amino acid PTP catalytic domain is made up of a twisted mixed β-sheet surrounded by α-helices. The active site is made up of several loops including the CxxxxxR-containing P-loop. There is no divalent cofactor. PTPs have no sequence or structural similarity with serine/threonine phosphatases. In topology and in overall structure, the classical PTP domain has no resemblance to the EYA-ED domain described above.

## 3. EYA Cellular Functions and Substrates

The structure of the EYA proteins cleanly separates individual biochemical activities by domains. The N-terminal domain houses the transactivation and putative threonine phosphatase activities, while the C-terminal domain houses the PTP activity. As a result, it is possible to link individual biochemical activities to cellular functions.

The cellular functions of the EYA proteins can be broadly divided into two categories: those originating from the transactivation role of EYA and those directly related to the phosphatase activities of EYA. In its transactivation role, a critical step is interaction with the SIX family of homeodomain proteins and subsequent translocation of EYA to the nucleus [18]. Mutations in the *Eya* and *Six* genes are associated with severe developmental defects in humans, mice, frogs, fish, and flies, speaking to the importance of the EYA-SIX transactivation function. Among the developmental processes regulated by the EYA-SIX complex are *Drosophila* eye development [11], neural plate development in xenopus [19], cochlear hair cell fate [20], and limb bud development [21]. These have all been reviewed elsewhere [8,10,22,23]. Here, we will focus on the cellular functions associated with the C-terminal tyrosine phosphatase (HAD) domain of the EYA proteins.

Elucidation of the contribution of the EYA tyrosine phosphatase activity to normal development is complicated by the fact that there are four *Eya* genes in vertebrates with some overlap in expression patterns. Furthermore, knockout strategies result in the loss of all EYA activities, making it difficult to specifically attribute a phenotype to the tyrosine phosphatase activity alone. There are two studies documenting a role for the EYA tyrosine phosphatase activity in normal development. We have shown that pharmacological inhibition of the EYA PTP activity delays vascular development in the post-natal mouse retina [24]. Curiously, while the PTP activity of the *Drosophila* Eya proteins was initially implicated in eye formation using GAL4-UAS mediated genetic rescue experiments [9,25], subsequent studies using genomic rescue showed that the tyrosine phosphatase activity was dispensable for fly eye formation and survival [26]. The emerging theme in *Drosophila* appears to be that, while the Eya PTP activity contributes to development, it is not essential [27]. A recent study suggests that fly Eya PTP activity contributes to the “robustness” of the retinal determination gene network; heterozygosity for other components of this network (*sine oculis* or *dachshund*) compromised the ability of PTP-dead Eya to rescue retinal defects [27]. In addition to a potential role for the eyes absent PTP activity in the aspects of eye development, there is some evidence (building upon an shRNA screen) that EYA1 PTP activity regulates sonic hedgehog (Shh) signaling during hind brain development [28].

The best understood cellular process in which EYA PTP activity plays a role is DNA damage repair. In eukaryotic cells, double strand break repair is initiated by a cascade of signaling events leading to the formation of a DNA damage repair complex (Figure 4a). One of the first steps in this process is the phosphorylation of Serine 139 of the minor histone protein H2AX to produce gamma-H2AX (γH2AX). H2AX is constitutively phosphorylated on its C-terminal tyrosine residue (Tyr142) by the WSTF kinase [29]. Upon DNA damage, the EYA PTP dephosphorylates Tyr142 of H2AX [30,31]. γH2AX dephosphorylated at Tyr142 permits the assembly of a DNA damage repair complex, while γH2AX phosphorylated at Tyr142 promotes apoptosis [30]. Hence, the EYA tyrosine phosphatase activity is critical in the repair versus apoptosis decision after DNA double-stranded breaks. Thus far, a role for the DNA damage repair function of the EYA proteins has been described in kidney development [30], developmental angiogenesis [24], and pathological angiogenesis [24,32,33]. Excitingly, the recognition that the EYA tyrosine phosphatase activity can contribute to disease pathologies has opened up the possibility of EYA PTP-targeted therapeutics. Several inhibitors of the EYA PTP activity have been identified [34,35], and some have been validated in animal models of disease (tumor growth and angiogenesis [32], pulmonary arterial hypertension [33], proliferative retinopathy [24]).

Other proposed substrates of the EYA PTP activity include estrogen receptor beta (ERβ; (Figure 4b) and the nine WD repeat protein WDR1 (Figure 4c). ERβ represses cell proliferation genes in the context of reproductive system development, neurogenesis, and pregnancy-related physiological changes, as well as acting as a potent tumor suppressor. ERβ was identified as an EYA2-binding protein in a yeast two hybrid screen [36]. ERβ is phosphorylated on Tyr36 by the c-ABL tyrosine kinase, making it transcriptionally active, able to promote coactivator recruitment to ERβ target promotors, and able to inhibit tumor growth in both cell culture and xenograft models. When EYA2 dephosphorylates ERβ at Tyr36, it reverses this tumor suppressor activity. WDR1 was identified as an EYA3 PTP substrate using a phospho-peptide array [37]. WDR1 promotes depolymerization of actin filaments decorated with the actin binding protein cofilin and promotes growth of various cancers (hepatocellular carcinoma, non-small cell lung cancer, breast cancer, and glioblastoma).

Multiple studies report that the N-terminal domain of the EYA proteins has threonine phosphatase activity across a range of substrates including MYC, NOTCH, and aPKCζ [12,13,14,15]. This activity was variously localized to the N terminus of EYA3 and EYA4, or to both the N-terminal domain and the ED domains. This has all been called into question by a study suggesting that interaction between EYA3 and PP2A is responsible for the threonine phosphatase activity previously attributed to the EYA3 N-terminal domain [16]. However, a more recent report failed to detect any PP2A during the EYA1-mediated dephosphorylation of Thr410 of aPKCζ [38]. This could possibly be ascribed to differences between EYA3 and EYA1. No doubt future investigations will shed further light on this issue.

## 4. Substrate Specificity among the EYA Proteins

The ED domains of the animal and plant EYA proteins have very high amino acid sequence identity. Notably, all of the residues implicated in catalysis are conserved [8]. Whether the substrate specificities of all EYA proteins are the same has not formally been investigated. An analysis of the amino acid sequences surrounding known substrate phospho-tyrosines revealed no clear motif [8]. Within the ED domain (hydrolase domain), it would be of interest to investigate the contribution (if any) of the cap domain to substrate specificity.

With a multi-domain protein such as EYA, it is possible that the N-terminal domain contributes to substrate recruitment and catalytic specificity. Such a mode of interaction has been described for other phosphatases (PP1 and calcineurin); a docking surface(s) distinct from the active site engages the substrate and confers specificity [39]. Furthermore, substrate specificity could also derive from protein–protein interactions. For the EYAs, interaction with the SIX family of transcription factors necessarily precedes nuclear translocation. Substrate specificity could thus arise from changes in sub-cellular localization. Indeed, for EYA to reach its nuclear target, H2AX, it would have to be translocated to the nucleus as a complex with a SIX protein. An EYA–SIX complex would then have to approach the H2AX substrate while it is wrapped in DNA. Thus, the EYA–SIX complex would be confronted by the electronegative phosphates of DNA. Recruitment to DNA via the SIX homeodomain could provide a mechanism through which the acidic EYA protein might be localized in proximity with this DNA-associated substrate. It is also possible that the intrinsic substrate specificities of the various EYA–SIX complexes differ from that of isolated EYA proteins. All of these aspects of EYA activity are yet to be investigated.

## 5. Evolutionary Origins of the EYA Tyrosine Phosphatase Domain

EYA orthologues have been reported across the animal kingdom [40], as well as in plants [41,42]. A domain enhanced lookup time accelerated (DELTA)-BLAST [43] restricted to fungi retrieved five hits of hypothetical proteins from different fungal species. The hit with the highest similarity score was a 672 amino acid protein from the anaerobic fungus *Pyromices finnis*, an obligate anaerobic chytrid fungus that grows in the gut of mammalian herbivores and is necessary for digestion of plant cell walls. Motifs I–III of the HAD phosphatases are present in the fungal ED, and motif IV is present in four out of five fungal EYAs (Figure 2b). A query of protozoa and yeast sequences returned no protein level similarities to the EYA domain, suggesting that it emerged when kingdom protista diverged into the three kingdoms of complex eukaryotes: plants, fungi, and animals (5) (Figure 2c). Indeed, an earlier evolutionary analysis suggested that the EYA domain, and most of the 22 HAD phosphatase families, were present in the choanoflagellate *Monosiga brevicolis* [4], commonly referred to as the last common ancestor of all metazoans [44]. We identified predicted EYA domains in at least two choanoflagellate species (Figure 5). Here too, motifs I–III are conserved, but motif IV is less conserved. Interestingly, choanoflagellates also have tyrosine kinases [2], suggesting the existence of phosphotyrosine-based signaling mechanisms.

Taken together, this examination of the origins of the EYA domain suggests that a proto-EYA likely diverged and evolved independently in each of the downstream metazoan kingdoms. The observations described above are also compatible with the EYA domain evolving independently of the other HAD phosphatases. In the absence of experimental characterization of early putative EYA orthologues, it remains a matter of speculation whether the early EYA domain containing proteins had tyrosine phosphatase activity. Pertinent to this discussion, it has been proposed that classical PTPs evolved before protein tyrosine kinases, and that this was driven by the ability of some promiscuous serine/threonine kinases to also phosphorylate tyrosine residues [45].

## 6. Conclusions and Perspectives

There is an expanding recognition that the eyes absent proteins play important roles in normal development and are important contributors to disease pathologies [8,10]. They also present the opportunity to examine the evolution of the large hydrolase superfamily as it acquired the ability to participate in the phosphotyrosine signaling machinery of metazoans. Much remains to be done. Outstanding questions pertaining to the EYA HAD domain include elucidation of the full repertoire of EYA tyrosine phosphatase substrates, whether the different EYA orthologues have distinct substrate specificities, and what contribution the N-terminal domain makes to the substrate specificity of the EYA proteins. It will also be interesting to see how future studies shape our understanding of the threonine-phosphatase activity in the N-terminal domain, and reveal the interplay between the transactivation, threonine phosphatase, and tyrosine phosphatase activities in this unusual member of the hydrolase super-family.

## Figures and Tables

**Figure 1 ijms-22-03925-f001:**
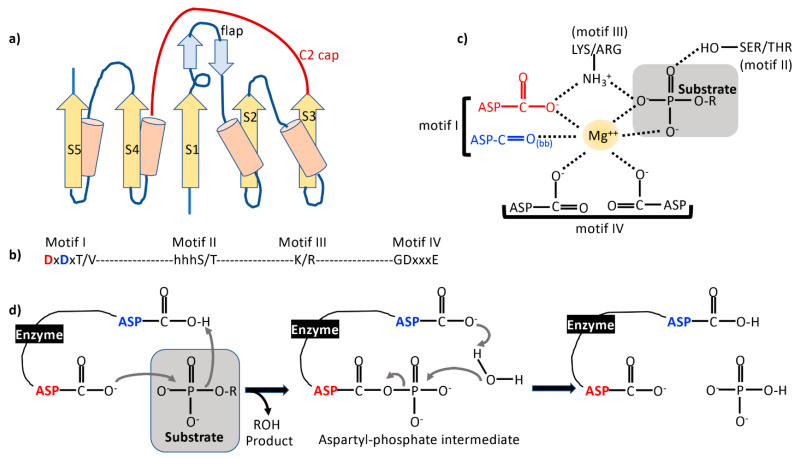
(**a**) Topology of a typical haloacid dehalogenase (HAD) phosphatase domain. β-strands are shown as yellow arrows and helices as cylinders. Strands are numbered S1–S5. Sample flap and C2 cap domains are illustrated. (**b**) The four sequence motifs that characterize HAD phosphatases. (**c**) Involvement of the amino acids in the conserved motifs in co-ordination of the obligate Mg^2+^ ion and substrate interaction. bb refers to a backbone carbonyl group. (**d**) Schematic of the HAD phosphatase reaction mechanism. The nucleophilic Asp (red) and the general acid-base Asp (blue) from motif I are color coded.

**Figure 2 ijms-22-03925-f002:**
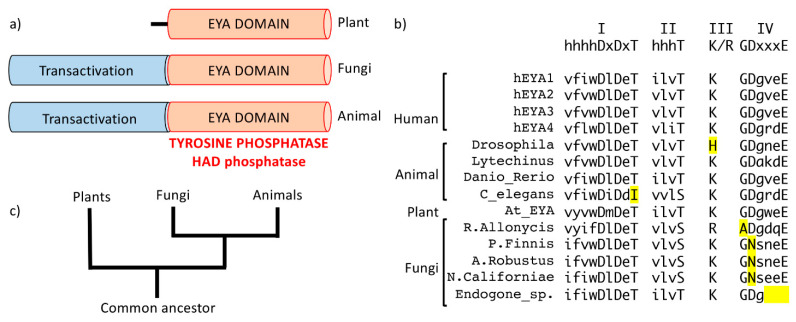
(**a**) Domain architecture of eyes absent (EYA) proteins from plants, fungi, and animals showing the conserved EYA domain (ED) in orange and the variable N-terminal (transactivation) domain in blue. Plants do not have an N-terminal domain. (**b**) The characteristic haloacid dehalogenase (HAD) motifs in various animal, plant, and fungal species. There is complete conservation of the two Asp residues in Motif I. Non-conserved residues are highlighted in yellow. One fungal species does not have the complete Motif IV. (**c**) An evolutionary tree showing the likely parallel evolution of EYA in the three kingdoms: plant, animal, and fungi.

**Figure 3 ijms-22-03925-f003:**
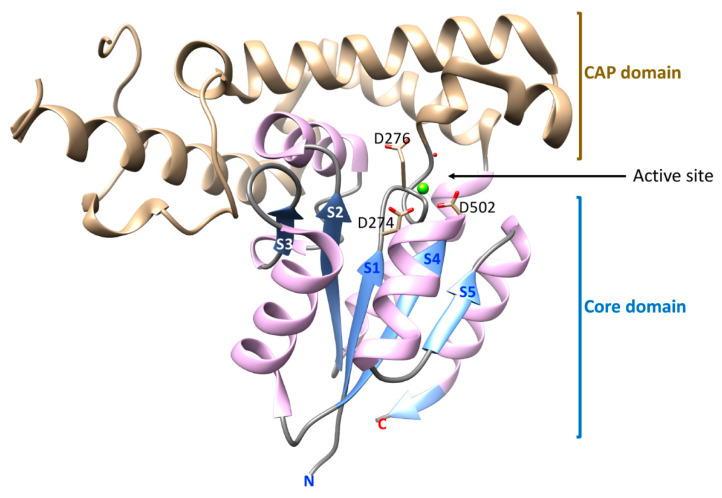
A ribbon drawing of the EYA2 ED domain generated using UCSF Chimera [17] from 3GEB.PDB [15]. The Mg^2+^ ion is depicted as a green sphere. The core and cap domains, as discussed here, are shown. β-strands that make up the Rossmanoid HAD core structure are numbered S1–S5, going from N- to C-terminus of the domain. Amino acid side-chains in the active site (the nucleophilic Asp and the genral acid-base Asp in Motif I, and the Asp in Motif IV that co-ordinates the metal ion) are shown in the active site.

**Figure 4 ijms-22-03925-f004:**
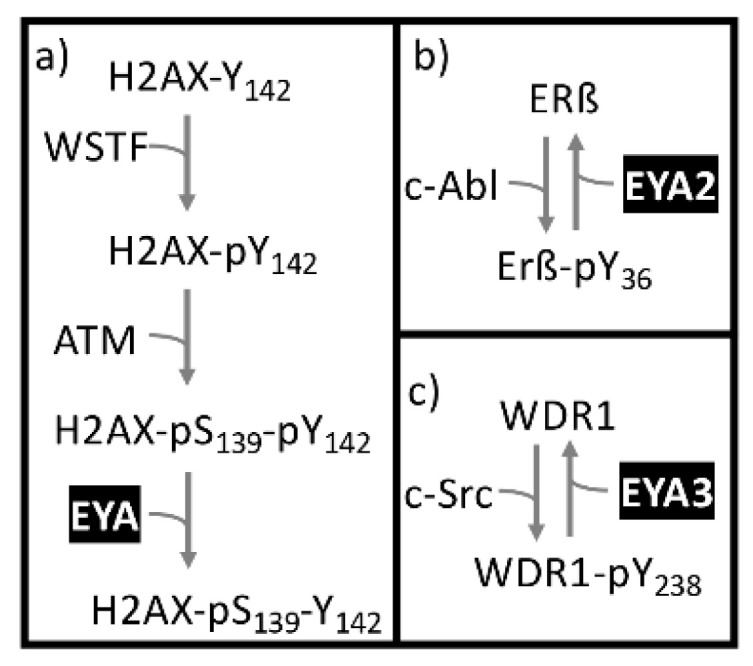
Known and proposed substrates of the eyes absent (EYA) protein tyrosine phosphatase (PTP) activity.(**a**) H2AX-Y142, (**b**) ERβ and (**c**)WDR1.

**Figure 5 ijms-22-03925-f005:**
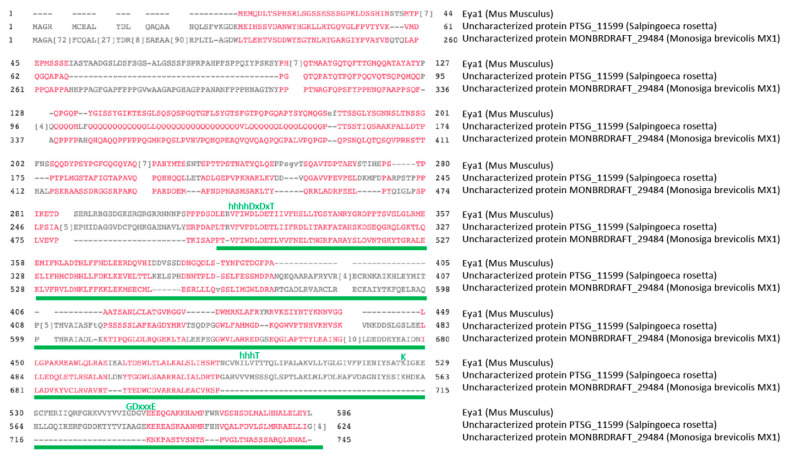
Amino acid sequence alignment of mouse Eya1 with the predicted proteins from two choanoflagellate species that contain a putative EYA domain. Characteristic HAD sequence motifs are shown above the alignment in green letters. The ED is indicated by the green line. BLAST searches were performed using www.ncbi.nlm.nih.gov/blast (accessed on 20 February 2021). domain enhanced lookup time accelerated BLAST(DELTA-BLAST) was used to find ED domains. Alignments were obtained using MULTALIGN (www.multalign.tolouse.fr) (accessed on 20 February 2021) and CLUSTAL Omega (https://www.ebi.ac.uk/Tools/msa/clustalo/) (accessed on 20 February 2021).

## Data Availability

Not applicable.

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
