# Peer review of "The Eyes Absent Proteins: Unusual HAD Family Tyrosine Phosphatases"

_ijms, 2021, doi:10.3390/ijms22083925_

Round 1

Reviewer 1 Report

Dear Editor,

I carefully read the manuscript by Roychoudhury and Hegde.

My comments and suggestions for the authors are the following:

  • The manuscript is overall well written though not very balanced in its parts. Authors should include in the paper a methods paragraph, detailing the search databases (e.g. PubMed, WOS, etc.) they consulted in writing the review and specifying the software used to generate the graphics and figures.
  • References should be prepared following the Instructions for the Authors of the Journal. Currently they are not in the appropriate style.

Author Response

Thank-you for the kind comments. Responses to specific suggestions are provided in italics below:

Methods for sequence alignment are provided in the legend to figure 5, page 10, and in the text. This being a review not involving any experimental work a specific Methods section is not included.

We have now modified the reference list to put Year after journal name in bold as indicated in the dot template provided by the journal.

Reviewer 2 Report

The review manuscript entitled “The eyes absent proteins: Unusual HAD family tyrosine phosphatases” provides a concise, latest and important review of structure and function of EYA proteins with HAD phosphatase activity. I read the manuscript with an interest, and I think that this theme may be had interest by experts on HAD phosphatases. On the other hand, I think that it may be difficult to be getting attention from many researchers. Minor concerns raised are shown as below. I think that these are important for the general reader's understanding of function of the EYA protein.

(1) Addition of explanations.

・Details of the phenotypes of the EYA-deficient Drosophila

・Relationship between mutations of the EYA and human genetic diseases

・Cellular distribution of the EYA protein in resting cells

・Description of the EYA proteins of organisms other than animals, plants and fungi

(2) Addition of Figures

・Function of the EYA proteins as transcriptional activators

・Function of the EYA proteins in double strand break repair

Author Response

Reviewer 2

We thank this reviewer for the detailed review. Our responses to individual comments are provided in italics below.

  • We thank Reviewer 2 for his/her kind words about this review regarding structure and function of EYA proteins with HAD phosphatase activity. We acknowledge that this review did not focus on the developmental roles of the EYA proteins. This is because this review is specifically about the HAD activity of the EYA proteins as solicited for a special issue on hydrolases. The role of the EYA proteins in development has been extensively reviewed elsewhere and reference to these reviews is provided on page 7. Phenotypes of Eya-deficient Drosophila and the relationship between mutations of EYA and human genetic diseases are likewise irrelevant to the topic under review.

・Cellular distribution of the EYA proteins in resting cells is also secondary to the topic of hydrolase structure and function. Nevertheless, it is discussed on page 9, lines 180 - 186 as it pertains to the HAD domain.

・Description of the EYA proteins of organisms other than animals, plants and fungi - Representative species of interest to the hydrolase activity are discussed starting on page 9, line 189.

(2) Addition of Figures -

・Function of the EYA proteins as transcriptional activators - The transcriptional activation function of the EYA proteins, while mentioned in the text for completeness, does not relate to the hydrolase domain. It is shown schematically in Figure 2.

・Function of the EYA proteins in double strand break repair - Since it pertains to the HAD domain of the EYA proteins we have added Figure 4 showing the various substrates of the PTP activity, including H2AX which is involved in DNA repair.

Reviewer 3 Report

This manuscript describes the function of EYA proteins as tyrosine phosphatases. Although a lot of researches remain to be explored, the role of EYA proteins with regard to the protein tyrosine phosphatase activity reported until now are explained. I recommend this manuscript for the publication in IJMS with a few minor corrections of editorial errors.

  1. 173: PTP-dead eya -> PTP-dead EYA
  2. 194: estrogen -> Estrogen
  3. 207: aPKCz -> Is this aPKCzeta? Then, use use Greek zeta

Author Response

We are very grateful to this reviewer for catching several typos in the document. Responses to each correction are provided in italics below.

  1. 173: PTP-dead eya -> PTP-dead EYA -        Here we are referring to Drosophila protein hence it is listed as “PTP-dead Eya”
  2. 194: estrogen -> Estrogen -     This has been corrected – thank-you!
  3. 207: aPKCz -> Is this aPKCzeta? Then, use use Greek zeta    -      This has been changed – thank-you!

Round 2

Reviewer 1 Report

Dear Editor,

I carefully read the revised version of the manuscript that is significantly improved compared to the previous one. I warmly recommend its acceptation in the Journal.